

# Impact of firework on nitrooxy-organosulfates in urban aerosols during Chinese New Year Eve

Qiaorong Xie[1,2], Sihui Su[1], Jing Chen[3], Yuqing Dai[4], Siyao Yue[2,5], Hang Su[5], Haijie Tong[6], Wanyu Zhao[2], Lujie Ren[1], Yisheng Xu[7], Dong Cao[8], Ying Li[2,9], Yele Sun[2], Zifa Wang[2], Cong-Qiang Liu[1], Kimitaka Kawamura[10], Guibin Jiang[8], Yafang Cheng[5], and Pingqing Fu[1*]

[1]Institute of Surface-Earth System Science, School of Earth System Science, Tianjin University, Tianjin 300072, China
[2]LAPC, Institute of Atmospheric Physics, Chinese Academy of Sciences, Beijing 100029, China
[3]School of Environmental Science and Engineering, Tianjin University, Tianjin, 300072, China
[4]School of Geography, Earth and Environmental Sciences, University of Birmingham, Birmingham, B15 2TT, UK
[5]Max Planck Institute for Chemistry, Multiphase Chemistry Department, Hahn-Meitner-Weg 1, 55128 Mainz, Germany
[6]Department of Civil and Environmental Engineering, The Hong Kong Polytechnic University, Hong Kong
[7]Chinese Research Academy of Environmental Sciences, Beijing 100012, China
[8]State Key Laboratory of Environmental Chemistry and Ecotoxicology, Research Center for Eco-Environmental Science, Chinese Academy of Sciences, Beijing 100085, China
[9]Department of Chemistry, California University, Irvine, California 92697-2025, United States
[10]Chubu Institute for Advanced Studies, Chubu University, Kasugai 487-8501, Japan
*Correspondence to*: Pingqing Fu (fupingqing@tju.edu.cn)

**Abstract.** Little is known about the formation processes of nitrooxy-organosulfates (nitrooxy-OSs) by nighttime chemistry. Here we characterize nitrooxy-OSs at a molecular level in firework-related aerosols in urban Beijing during Chinese New Year. High-molecular-weight nitrooxy-OSs with relatively low H/C and O/C ratios and high unsaturation, which are potentially aromatic-like nitrooxy-OSs, considerably increased during the New Year's Eve. We find that large quantities of carboxylic-rich alicyclic molecules possibly formed by nighttime reactions. The sufficient abundance of aliphatic-like and aromatic-like nitrooxy-OSs demonstrates that both anthropogenic and biogenic volatile organic compounds are essential precursors of urban secondary organic aerosols (SOA). Besides, more than 98% of nitrooxy-OSs were extremely low-volatile organic compounds that could easily partition into and consist in the particle phase, and affected the volatility, hygroscopicity, and even toxicity of urban aerosols. Our study provides new insights into the formation of nitrooxy-organosulfates from anthropogenic emissions through nighttime chemistry in the urban atmosphere.

## 1 Introduction

Secondary organic aerosols (SOA) are essential components in atmospheric aerosols that are related to climate change, air quality and human health. They are generated through not only daytime photooxidation but also nighttime chemical oxidation from both biogenic and anthropogenic volatile organic compounds (VOCs) (Hallquist et al., 2009; Rollins et al., 2012; Nozière et al., 2015; Huang et al., 2019). Nitrooxy-organosulfates (nitrooxy-OSs) with nitrooxy ($-ONO_2$) and sulfate ester groups ($-OSO_3H$) (Surratt et al., 2008; Lin et al., 2012) substantially participate in the formation of SOA (Tolocka and





Turpin, 2012; Ng et al., 2017; Bruggemann et al., 2020). Moreover, nitrooxy-OSs can alter the surface hygroscopicity of aerosol particles because of their water-soluble and fat-soluble properties, promoting the production of cloud condensation nuclei (Schindelka et al., 2013) and also increasing the light absorption of organic aerosols (Nguyen et al., 2012).

Nitrooxy-OSs can be generated within both biogenic (Iinuma et al., 2007b; Surratt et al., 2007; Gómez-González et al., 2008;

Surratt et al., 2008) and anthropogenic SOA (Tao et al., 2014; Riva et al., 2015). The main precursors of biogenic nitrooxy-OSs were isoprene, monoterpenes, sesquiterpenes, and aldehyde, as previous studies observed biogenic nitrooxy-OSs in the isoprene chamber experiments (Gómez-González et al., 2008; Surratt et al., 2008), a forest (Iinuma et al., 2007b) and urban aerosols (Lin et al., 2012). Compared to biogenic nitrooxy-OSs studies, few research activities has been carried out focusing on the anthropogenic nitrooxy-OSs. Tao et al. (2014) found that long-chain alkenes from traffic emissions are possible

precursors of long-chain alkyl nitrooxy-OSs in urban aerosols in Shanghai. A recent study reported the presence of nitrooxy-OSs in the polar regions (Ye et al., 2021).

The formation of nitrooxy-OSs due to nighttime chemistry is less understood so far. Surratt et al. (2008) suggested that nitrooxy-OSs can be formed from the combination of organonitrates and sulfates under acidification, while organonitrates are preferably produced by nighttime $NO_3$ radical oxidation than daytime photooxidation (Rollins et al., 2012; Kiendler-

Scharr et al., 2016; Huang et al., 2019). Iinuma et al. (2007) reported that some monoterpene nitrooxy-OSs (e.g., $C_{10}H_{17}NO_7S$, $C_{10}H_{18}N_2O_7S$, and $C_5H_{10}N_2O_{11}S$) were only detected in nighttime aerosols, indicating the importance of $NO_3$ radicals in the nighttime chemistry. Fireworks are frequently conducted as traditional activities to celebrate popular festivals in particular with the New Year's Eve, emitting large quantities of pollutants into the atmosphere (Vecchi et al., 2008; Huang et al., 2012). It is found that lots of semi-volatile to volatile organic compounds, such as $n$-alkanes and polycyclic aromatic

hydrocarbons (PAHs), released during firework-related events (Sarkar et al., 2010; Feng et al., 2012), can be essential precursors of anthropogenic nitrooxy-OSs in aerosols (Tao et al., 2014; Riva et al., 2015). Even though, the knowledge of chemical and physical behaviors of nitrooxy-OSs in firework-related urban aerosols is very sparse, particularly for high-molecular-weight (HMW) compounds because of their molecular complexity.

To fill this research gap, the molecular characterization of HMW nitrooxy-OSs in firework-related aerosols during nighttime

is reported in this study based on the measurements from Fourier transform ion cyclotron resonance mass spectrometry (FT-ICR MS) with ultrahigh resolution and mass accuracy. FT-ICR MS has been proven to be a powerful tool to reveal the complicated organic matter in environmental samples at a molecular level (Dzepina et al., 2015; Qi et al., 2020; Qi et al., 2021). Our study presents elemental compositions, classifies the organic mixtures into different categories to identify potential origins of nitrooxy-OSs and to investigate their possible chemical structures and precursors. The volatility of

different nitrooxy-OSs is predicted and discussed as well.



## 2 Experimental section

### 2.1 Aerosol sampling

Daytime/nighttime aerosol samples (n=6) were sampled from 21$^{st}$ to 23$^{rd}$ of January 2012 in an urban site at the Institute of Atmospheric Physics, Chinese Academy of Sciences (39°58′28″N, 116°22′13″E), Beijing. The samples include NYE D

(New Year's Eve daytime before the fireworks), NYE N (New Year's Eve nighttime during the fireworks), LNY D (lunar New Year's Day daytime after the fireworks), LNY N (lunar New Year's Day nighttime), Normal D (Normal Day daytime), and Normal N (Normal Day nighttime) (Xie et al., 2020b). Detailed sample information can be found in Table S1. The total suspended particle (TSP) samples were collected on prebaked quartz filters (Pallflex) using a high-volume aerosol sampler, and then were refrigerated at −20°C until analyzation. Field blank filters were collected following the same procedure.

Besides, two-day air mass backward trajectories show that the air was primarily originated from the clean northwest region during the sampling period.

### 2.2 FT-ICR MS analysis

The method to extract water-soluble organic carbon (WSOC) fractions from each aerosol samples were taken from our previous studies (Xie et al., 2020a; Xie et al., 2020b). After extracted with ultrapure Milli-Q water, WSOC fractions were

eluted from a solid-phase extraction cartridge (Oasis HLB, Waters, USA) using methanol, and were analyzed using a 15.0 T Bruker Solarix FT-ICR MS (Bruker Daltonik, GmbH, Bremen, Germany) with the negative ESI ionization mode. The detection mass ranges were 180−1000 Da. The mass spectra were internally calibrated via the Data Analysis software. The mass accuracy was within 1ppm and the peaks of signal to noise ratio higher than 6 were assigned for further analysis.

The formulae containing C, H, N, O, and S atoms, namely CHONS compounds, in WSOC fractions of urban aerosols were

assigned from the ESI FT-ICR MS. The number and the total intensity of CHONS compounds attributed to 14–29% and 10–28% of the total assigned compounds, respectively (Figure S1). As the predominant fraction occupying more than 85%, CHONS compounds with O/S ≥ 7 were tentatively deemed to be nitrooxy-OSs in present work, supporting the assignment of a –OSO$_3$H group and a –ONO$_2$ group in molecules (Kuang et al., 2016; Wang et al., 2016). However, other sulfur-containing compounds (e.g. sulfonates) might also be introduced and impact the analyzation due to lack of the structure information of

ions from tandem MS experiments (El Haddad et al., 2013; Riva et al., 2015).

## 3 Results and discussion

### 3.1 General molecular characterization of nitrooxy-OSs

Table 1 shows that slightly higher compounds in number frequency was found in normal nighttime samples (1094 in Normal N) than in normal daytime (885 in Normal D), which was in agreement with previous studies (Pinxteren et al., 2009; O'Brien

et al., 2014; Wang et al., 2016). However, 690 nitrooxy-OSs were observed in NYE D before the firework periods, while it



considerably increased up to 2050 in NYE N during the firework periods, indicating a pronounced increment at night during the firework event. This can be explained by significant precursors emitted from fireworks (Kong et al., 2015) that produce nitrooxy-OSs via the nighttime $NO_3$ radical chemistry (Riva et al., 2015). Meanwhile, the heavy emissions of nitrogen oxide during the fireworks event could elevate the production rate of $NO_3$ radicals (Ljungström and Hallquist, 1996; Kiendler-

Scharr et al., 2016), and previous study showed a good correlation between $NO_3$ and the total concentration of nitrooxy-OSs at the night (Nguyen et al., 2014).

Different from the detected compounds in NYE D, here were 1411 nitrooxy-OSs that were only detected in NYE N, which contributed 69% of total number of nitrooxy-OSs in the sample (Figure 1). Moreover, their relative intensities also accounted for nearly half of that in NYE N. These results indicate that extensive burning of firecrackers offered many specific

precursors for the formation of new nitrooxy-OSs.

Nitrooxy-OSs were identified as $N_1O_7S_1$–$N_1O_{13}S_1$ and $N_2O_7S_1$–$N_2O_{14}S_1$ species (Figures 2 and S2). Here, $N_1O_7S_1$ compounds refer to formulae containing one nitrogen, seven oxygen, and one sulfur elements, and so for the other species. Their number concentrations decrease with the increase of oxygen content in molecules. $N_1O_nS_1$ species are the predominant nitrooxy-OSs, the number and the intensity of $N_1O_nS_1$ species occupied 65–82% and 62–80% of the total detected nitrooxy-

OSs, respectively. During firework periods, up to 1300 species of $N_1O_nS_1$ were detected in NYE N, twice as many as other samples. The effect of pyrotechnics on nitrooxy-OSs become more substantial with the increased oxygen atom number (Figure 2). Similarly, the intensity of $N_1O_nS_1$ species doubled in NYE N compared to other samples. $N_2O_nS_1$ species may have other nitrogenous functional groups (e.g., amino and nitro groups) in addition to nitrate functional group. During non-firework periods, there were an average of 300 species of $N_2O_nS_1$, but the number increased to 724 in NYE N. Moreover, the

contribution of fireworks to $N_2O_nS_1$ species was higher than that of $N_1O_nS_1$ compounds, possibly because some released amino acids and their derivatives react to form nitrooxy-OSs with two nitrogen-containing functional groups.

Table 1 and Figure S3 present the arithmetic and weighted mean elemental ratios of total nitrooxy-OSs for each sample, respectively. The average molecular weights rose from $411 \pm 69$ Da (Normal D) to $417 \pm 78$ Da (Normal N) and from $398 \pm 69$ Da (NYE D) to $449 \pm 93$ Da (NYE N). The average molecular formulae are $C_{17}H_{25}O_{8.5}N_{1.2}S_{1.0}$ and $C_{18}H_{24}O_{8.6}N_{1.3}S_{1.0}$ for

Normal D and Normal N, and $C_{17}H_{24}O_{8.6}N_{1.1}S_{1.0}$ and $C_{21}H_{26}O_{9.1}N_{1.4}S_{1.0}$ for NYE D and NYE N, respectively. Nitrooxy-OSs in firework-related aerosols had relatively higher C and O contents, indicating that many HMW compounds had a higher extent of oxidation. Moreover, both O/C and H/C ratios of nitrooxy-OSs lessened in NYE N, along with increases of unsaturation parameters of double-bone equivalent (DBE) values and DBE/C ratios. Similar trends were found for the intensity-weighted average elemental ratios of compounds with high $DBE_w$ values, but low $O/C_w$ and $H/C_w$ ratios (Figure

S3). Compared with other studies (Jiang et al., 2016; Lin et al., 2012b), our results suggested that there were more aromatic compounds in aerosol samples. Additionally, lots of nitrooxy-OSs with high DBE values ($\geq 7$) were only detected in NYE N (Figure 3). They were mostly located in the region of aromatic index (AI) higher than 0.5, referring to their condensed aromatic ring structure. From Table 1, 21 and 38 compounds with AI > 0.5 were observed in Normal D and Normal N,





respectively. Compared with 16 these compounds in NYE D, there were up to 83 compounds in NYE N. These results possibly indicate that pyrotechnic emissions have substantial impacts on the formation of aromatic-like nitrooxy-OSs.

**3.2 Van Krevelen diagram division**

The Van Krevelen (VK) diagram is widely applied to depict the evolution of organic mixtures and to identify possible
origins of organic aerosols by differentiate major known categories of the natural and anthropogenic organic matter (Noziere et al., 2015; Bianco et al., 2018). Here, we applied the VK to investigate nitrooxy-OSs in firework-related aerosols. The particularly seven classification areas are shown in Figures 3 and S4, and their stoichiometric ranges are displayed in Table S2. The two most populated regions correspond to carboxylic-rich alicyclic molecules (CRAMs-like)/lignin-like (49−66% and 40−49% of the total number and intensity) and aliphatic/peptides-like (21−33% and 24−38% of the total number and
intensity) classes, which followed by carbohydrates-like (6−12% and 11−19% of total number and intensity) and tannins-like (4−7% of total number and intensity) classes. It is found that more than 98% of nitrooxy-OSs belong to these four categories. Previous studies reported that the majority of WSOC fractions were lignin-, lipids-, and aliphatic/peptides-like classes in aerosols and cloud water (Wozniak et al., 2008; Zhao et al., 2013; Bianco et al., 2018). However, nitrooxy-OSs in the present work had relatively high O/C ratios, potentially because they were consistent with covalently bound $HSO_4^-$ (Romero and
Oehme, 2005) in HMW sulfur-containing compounds (Wozniak et al., 2008). Besides, nitrooxy-OSs also had high H/C ratios, indicating that sulfation, nitration, or functionalization processes led to mostly saturated compounds.

Although the number of all seven types of nitrooxy-OSs increased obviously in NYE N compared with other samples, the impacts of fireworks on nitrooxy-OSs are different in categories. As for the most abundant CRAMs-like nitrooxy-OSs, they were more abundant during the nighttime (620 in Normal N) than daytime (375 in Normal D). However, NYE N contained
about 1354 CRAMs-like compounds, which were three times more than NYE D. The relative contribution of the number of these compounds was 60% in NYE N, substantially higher than that in NYE D (45%). The total intensity of such compounds in NYE N was about eight times higher than those in NYE D. These observations demonstrate that nighttime oxidation is important in the formation of CRAMs-like nitrooxy-OSs, especially in the presence of abundant firework-related precursors. CRAMs contain the structures of carboxylated alicyclic and large and fused nonaromatic rings with a high ratio of
substituted carboxyl groups (Bianco et al., 2018). However, some CRAMs-like nitrooxy-OSs located in the aromatic area (AI > 0.5). Thus, there may exist some aromatic-like compounds with some degree of alkylation that have been mistaken for nonaromatic class (Kourtchev et al., 2016; Tong et al., 2016). Also, it was worth noting that nitrooxy-OSs of aromatic region could be lignin-like compounds, which contains aromatic rings in their chemical structures. The results are consistent with the high unsaturation of compounds in the region as described above.
Aliphatic/peptides-like nitrooxy-OSs have low DBE values and H/C ratios, indicating high degree of saturation and long carbon lengths. Aliphatic-like nitrooxy-OSs (e.g. acyclic compounds) are mainly derived from alkanes, alcohols, ethers, ketones, aldehydes, esters, et al., from anthropogenic and natural emissions. Peptides-like nitrooxy-OSs are primarily derived from functionalized amino acids, peptides, and protein fragments. Unlike CRAMs-like compounds, average relative



contributions of both the number and the total intensity of aliphatic/peptides-like nitrooxy-OSs were more abundant during daytime (~40%) than nighttime (~30%). However, there are 427 species of aliphatic/peptides-like nitrooxy-OSs in NYE N, which is twice as many as other samples. The intensity of each compound was also higher in NYE N with bigger symbol sizes (Figure 3b), and the total intensity was about three times higher than other samples. These results demonstrate the

importance of anthropogenic precursors for the formation of aliphatic/peptides-like nitrooxy-OSs, though they could be more susceptible to photochemical reactions than nighttime chemistry from biogenic precursors.

Contrary to CRAMs-like, the carbohydrates-like nitrooxy-OSs with high intensity have high H/C and O/C ratios and saturation, indicating highly oxidized and alkylated. Compared with other reported sulfur-free compounds (Wozniak et al., 2008; Bianco et al., 2018), the abundance of carbohydrates-like nitrooxy-OSs increased significantly. The increase is

reasonable because carbohydrates and their derivatives are polyhydroxy aldehydes, polyhydroxy ketones and organic compounds, which can be hydrolyzed to form polyhydroxy aldehydes or polyhydroxy ketones, tending to generate OSs and nitrooxy-OSs. Moreover, $N_2O_nS_1$ species were more abundant than $N_1O_nS_1$ species (Figure S5), indicating a trend toward easily functionalization. Although the number of carbohydrates-like compounds in NYE N was similar to other samples, the intensity of them increased, potentially indicating an increase of the concentration of them. As for tannins-like classes, which

are also highly oxygenated organic compounds (Bianco et al., 2018), their content in NYE N was also more abundant than other samples.

Considering the lipids-like, unsaturated hydrocarbons and aromatic structures classes, only less than 1.5% compounds are in these regions. Lipids-like organics, containing monoglycerides, diglycerides, fats, fat-soluble vitamins, and sterols, primarily originate from biogenic materials and phospholipids (Gurganus et al., 2015; Bianco et al., 2018), but most nitrooxy-OSs are

of secondary origin. Unsaturated hydrocarbons compounds are mostly composed of carbon and hydrogen atoms, while nitrooxy-OSs have lots of heteroatoms. As for aromatic compounds, they are mainly produced by combustions as the indicators of anthropogenic origin. The limited number of nitrooxy-OSs detected in this region may be due to the tendency of some alkylated compounds to fall into other categories (e.g., CRAMs-like).

### 3.3 Subgroups and potential precursors

Figure S6 showed that compared with NYE D, nitrooxy-OSs (> 1500 compounds) were densely distributed with high DBE values (≥ 7) during the fireworks event, especially for those in the HMW region. Most of the nitrooxy-OSs were aromatics with $X_c$ higher than 2.5 (Yassine et al., 2014) and lower O/C (≤ 0.5) and H/C (≤ 1.5) ratios. Besides, high intensities of these highly unsaturated compounds indicated their sufficient contents in aerosols. From Figures 4 and 5, the DBE values and C numbers of $N_1O_nS_1$ species of nitrooxy-OSs in NYE N varied separately within the range of 0–23 and 6–35, which were

upper than the average value of DBE (0–16) and C number (6–27) in other samples. Although the abundance of nitrooxy-OSs of DBE (4–10) and C number (10–20) in the nighttime was higher than that during the daytime, the number of nitrooxy-OSs with DBE and C number in the range of 4–18 and 10–20 was even higher in the NYE N. These highly unsaturated nitrooxy-OSs are aromatics, which may be originated from firework-related aromatic VOCs or PAHs (Riva et al., 2015).



Nitrooxy-OSs have an extensive range of unsaturation with DBE values ranging from 0 to 23 (Figure 4). Previous studies have reported that nitrooxy-OSs (e.g. $C_{10}H_{17}NO_7S$ (1), $C_9H_{15}NO_8S$ (2)) can be formed from biogenic VOCs (e.g. α-pinene and limonene) (Iinuma et al., 2007a; Surratt et al., 2008; Cai et al., 2020). $C_{10}H_{17}NO_7S$ and $C_9H_{15}NO_8S$ (Figure 4) have the same degree of unsaturation as their precursors and show the strongest intensity among all nitrooxy-OSs, which demonstrate

that α-pinene and limonene are the primary precursors of biogenic nitrooxy-OSs. A continuous series of corresponding family series was also detected in firework-related aerosols, namely $C_nH_{2n-3}NO_7S$ (n=9–22) and $C_nH_{2n-3}NO_8S$ (n=9–24) (Figure 5).

The nitrooxy-OSs were divided into three main categories to illustrate the molecular difference. Group A comprises aliphatic-like nitrooxy-OSs (DBE ≤ 2), which is featured by long alkyl carbon chains with high saturation; Group B contains

aromatic-like nitrooxy-OSs ($X_c > 2.5$) with high unsaturation; Group C represents the rest compounds similar to biogenic nitrooxy-OSs with a moderate extent of saturation. As for the aliphatic-like nitrooxy-OSs, such as $C_{18}H_{35}NO_9S$ (3) and $C_{12}H_{25}NO_7S$ (4) (Figure 4), they have saturated and long carbon chains, which may source from precursors such as long-chain alkenes, alkanes, and fatty acids by photooxidation. Both had relatively high intensities and consecutive family series i.e. $C_nH_{2n-1}NO_9S$ (n=9–22) and $C_nH_{2n+1}NO_7S$ (n=6–24), respectively (Figure 5). These precursors may also produce nitrooxy-

OSs under high $NO_x$ conditions or break double bonds to form intermediate products through photooxidation, and then form nitrooxy-OSs with sulfates. Some possible formation mechanisms are proposed in Figure 4(g). It is noted that, except for compounds (1) and (2), the intensity of each aliphatic-like nitrooxy-OSs are higher than others in daytime samples, highlighting the importance of photooxidation to their generation.

The number of the aromatic-like nitrooxy-OSs were the most abundant among all measured nitrooxy-OSs. Figures S6(d) and

S8(h) showed that there were slight differences between NYE D and NYE N for aliphatic and biogenic nitrooxy-OSs. Nonetheless, compared with the daytime, the number of aromatic-like nitrooxy-OSs considerably enhanced in NYE N in particular with the HMW ones. Riva et al. (2015) using side by side comparison experiments proved that the generation of OSs and sulfonates from PAHs was enhanced with the acidified sulfate seed aerosols existed. Their results implied that aromatic-like nitrooxy-OSs might efficiently be generated through PAHs and sulfate ions released from fireworks at night

without the participation of photochemistry. For instance, the aromatic-like nitrooxy-OSs with multiple benzenes can be generated from carboxyl compounds, which is a oxidation products of pyrene (Juhasz and Naidu, 2000). Some possible structures, such as $C_{18}H_{15}NO_{11}S$ (5), $C_{16}H_{13}NO_9S$ (6), and $C_{11}H_{15}NO_8S$ (7), and proposed formation mechanisms are displayed in Figures 4, respectively. Their corresponding family series (i.e. $C_nH_{2n-21}NO_{11}S$ (n=17–28), $C_nH_{2n-19}NO_9S$ (n=15–29), and $C_nH_{2n-7}NO_8S$ (n=8–29)) have more carbon atoms than the biogenic and aliphatic nitrooxy-OSs (Figure 5), possibly

because they are formed via the polymerization process or derived from aromatic compounds with high carbon content. Besides, it is noted that the intensity of each aromatic-like nitrooxy-OSs was lower than the other two groups and decreased with the increase of unsaturation (Figures 4 and Table S3). These may because the water solubility of aromatic-like nitrooxy-OSs decreases with the increasing unsaturation. However, these nitrooxy-OSs are possibly present in the non-WSOC fractions, which requires further investigations.



### 3.4 Volatility characteristics and molecular corridors

Molecular corridors that constrained by two boundary lines of sugar alcohols $C_nH_{2n+2}O_n$ with O/C = 1 and linear *n*-alkanes $C_nH_{2n+2}$ with O/C = 0 are used for a better understanding of the chemical and physical properties in SOA evolution (Shiraiwa et al., 2014; Li et al., 2016; Shiraiwa et al., 2017). From Figure 6(a-b), more than 98% of nitrooxy-OSs detected in present study were located in the region of extremely low-volatile organic compounds (ELVOC) with saturation mass concentration ($C_0$) < $3\times10^{-4}$ µg m$^{-3}$ (Donahue et al., 2011; Murphy et al., 2014) and a molar mass higher than 250 g mol$^{-1}$. Moreover, the volatility varies among $N_1O_nS_1$ and $N_2O_nS_1$ species because of the differences in their molecular composition and structures. Compared with $N_1O_nS_1$, many of the $N_2O_nS_1$ species have lower volatility and higher O/C ratio, nearing the sugar alcohols $C_nH_{2n+2}O_n$ line. Lots of these compounds with higher intensity were found in NYE N than other samples, which suggests that highly oxidized compounds are produced in large quantities via nighttime chemistry after firework emissions. Moreover, the volatility of nitrooxy-OSs potentially decreased with the increase of molecular weight and unsaturation. Compared with compounds in NYE D, numbers of nitrooxy-OSs with low volatility were only detected in NYE N, possibly because of the increase of HMW aromatic-like nitrooxy-OSs affected by firework emissions. Besides, it was worth noting that the volatility of nitrooxy-OSs was lower than that of OSs (Xie et al., 2020a), possibly because they are highly functionalized compounds.

The molecular corridor includes three primary parts, consisting of low, intermediate, and high O/C ratio corridors (LOC, IOC, and HOC) (Shiraiwa et al., 2014). The plentiful gas-phase oxidation products of alkanes fall into LOC, which near the alkanes line. Conversely, the aqueous-phase reaction and autoxidation products were in HOC, closing to the sugar alcohols line. The IOC corridor is the area connecting LOC and HOC. Nitrooxy-OSs observed in this work are dominantly found in the IOC molecular corridors. $N_1O_nS_1$ are closer to the LOC corridors, while $N_2O_nS_1$ are closer to the HOC corridors. These results suggest that most of firework-related nitrooxy-OSs were possibly gas- and/or particle-phase autoxidation or dimerization products. For instance, nitrooxy-OS can be formed through hydroxynitrate gas-phase products reactively up taking onto acidified sulfate seed aerosols through the esterification of the hydroxyl group with sulfuric acid (Surratt et al., 2008). Jay and Stieglitz (1989) also found hydroxynitrates produced by the oxidation of α-pinene induced by NO$_3$ at night. Also, several newly firework-related nitrooxy-OSs with higher molecular weight were possibly generated from dimerization and oligomerization in the particle phase. These results demonstrate that the dimerization and functionalization of nitrooxy-OSs can be substantially enhanced in the particle-phase with rising pollutant concentrations and varying reaction scenarios.

Although nitrooxy-OSs have been frequently reported within aerosols (Lin et al., 2012; O'Brien et al., 2014; Cai et al., 2020), deposited sediment (Zhang et al., 2016), and atmospheric water such as cloud water (Zhao et al., 2013), rain (Altieri et al., 2009), or fog (Mazzoleni et al., 2010), they were different from those in the present work because of the unique chemical reactions during the fireworks (Figure 6c). Organics in atmospheric water with aqueous phase reactions are highly oxidized and close to the sugar alcohol line. Although a fraction of cloud-water nitrooxy-OSs overlap with those in aerosols, the firework-related nitrooxy-OSs in our work showed a higher molar mass and lower volatility than urban aerosols reported



previously (Lin et al., 2012; O'Brien et al., 2014), especially at NYE night, potentially because of increased dimerization and oligomerization reactions.

## 4 Summary and Prospective

The present study provides unique information about the important contributions of anthropogenic precursors, as well as biogenic precursors, to the formation of nitrooxy-OSs in ambient aerosols during the firework events. Instead, numbers of nitrooxy-OSs were potentially derived from alkene, fatty acids, and aromatics, and their derivatives compared to biogenic-related nitrooxy-OSs. The surfactant properties of ambient aerosol particles may be influenced after coupling with hydrophilic functional groups of nitrooxy and sulfate, which affect the formation of cloud condensation nuclei. Furthermore, influenced by the fireworks emission, a lot of organonitrates in the gas-phase can partition into the particles phase by forming nitrooxy-OSs with low volatility as ELVOC, thus participating in the organic nitrogen cycle. Besides, they also affect the $NO_x$ cycle in the atmosphere. Our results highlight that fireworks emission considerably contributes to the formation of nitrooxy-OSs and will have an important influence on atmospheric physical and chemical processes. Nevertheless, nighttime chemistry of $NO_3$ radicals were substantially involved in the generation of nitrooxy-OSs, particularly for aromatic-like compounds. Such complex mechanisms warrant further investigations.

*Data availability.* The dataset for this paper is available upon request from the corresponding author (fupingqing@tju.edu.cn).

*Competing interests.* The authors declare that they have no conflict of interest.

*Acknowledgements.* This work was funded the National Natural Science Foundation of China (grant nos. 41625014 and 41961130384).

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



**Table 1.** The concentrations of chemical components and the number and elemental characteristics of nitrooxy-OSs in the Beijing aerosol samples.

| Sample ID | NYE D | NYE N | LNY D | LNY N | Normal D | Normal N |
|---|---|---|---|---|---|---|
| WSOC ($\mu$gC m$^{-3}$) | 5.1 | 11.6 | 3.8 | 5.1 | 2.8 | 2.7 |
| Number frequency | 690 | 2050 | 1097 | 1113 | 885 | 1094 |
| Molecular weight (Da) | 398±69 | 449±93 | 414±76 | 408±70 | 411±69 | 417±78 |
| O/C | 0.48±0.18 | 0.42±0.16 | 0.45±0.17 | 0.45±0.19 | 0.44±0.20 | 0.45±0.18 |
| H/C | 1.43±0.36 | 1.29±0.37 | 1.35±0.39 | 1.37±0.38 | 1.41±0.36 | 1.37±0.35 |
| OM/OC | 2.01±0.31 | 1.89±0.29 | 1.96±0.31 | 1.96±0.33 | 1.94±0.34 | 1.94±0.31 |
| DBE | 6.51±3.49 | 9.22±4.69 | 7.93±4.34 | 7.56±4.01 | 7.17±3.69 | 7.55±4.16 |
| DBE/C | 0.38±0.18 | 0.44±0.18 | 0.42±0.18 | 0.41±0.18 | 0.39±0.17 | 0.42±0.17 |
| Compound number | | | | | | |
| AI=0 | 429 | 791 | 553 | 624 | 500 | 610 |
| 0<AI<0.5 | 524 | 1955 | 963 | 1020 | 799 | 1398 |
| 0.5≤AI<0.67 | 16 | 81 | 22 | 23 | 21 | 38 |
| 0.67≤AI | 0 | 2 | 0 | 0 | 0 | 0 |




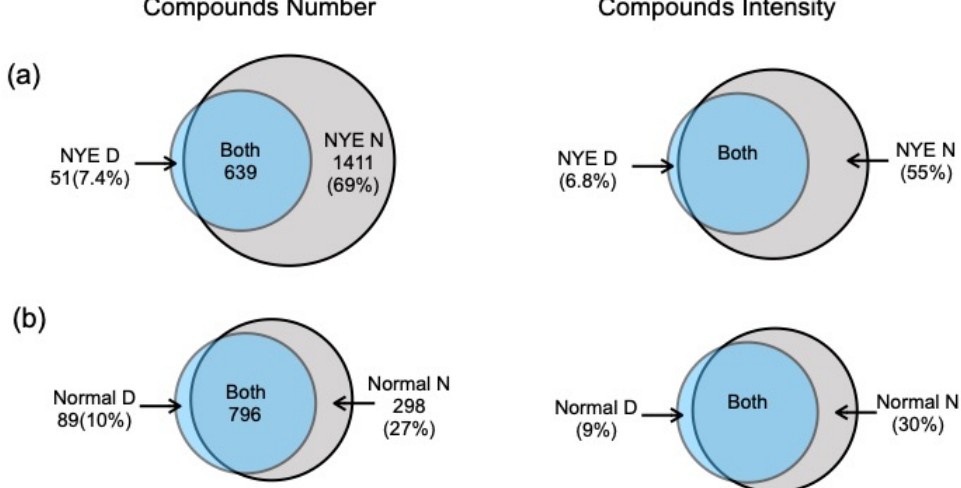

**Figure 1.** The number and intensity of nitrooxy-OSs compounds in (a) samples NYE D and NYE N and (b) samples Normal D and Normal N. The common compounds in both daytime and nighttime samples present as both with a number. The percentage numbers represent the proportion of the unique ones in the total nitrooxy-OSs of each sample.





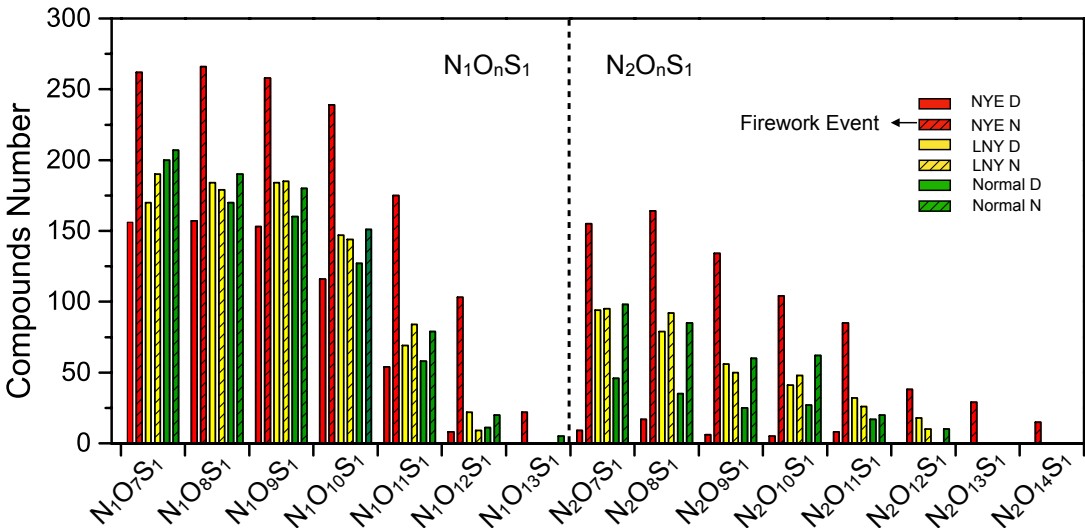

**Figure 2.** Classification of nitrooxy-OSs according to the numbers of N, S and O atoms in their molecules.





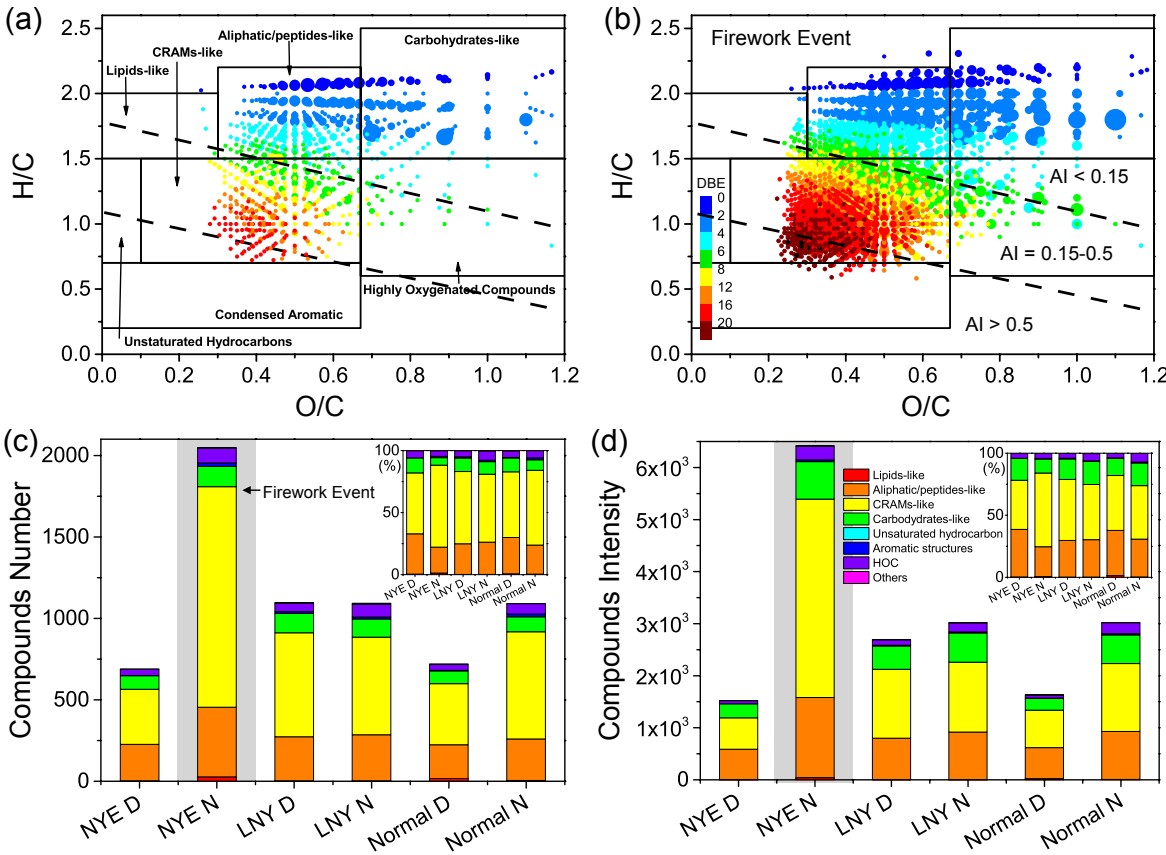

**Figure 3.** Typical Van Krevelen symbols for (a) NYE D and (b) NYE N. Black dotted lines show various AI value ranges and black lines denote class identification. The size of the plots represents the relative intensities of nitrooxy-OSs on a logarithmic scale. The colored bars of (a) and (b) reflects the DBE values. Bar diagrams of (c) and (d) show the number and intensity contribution of major classes in different samples, respectively.



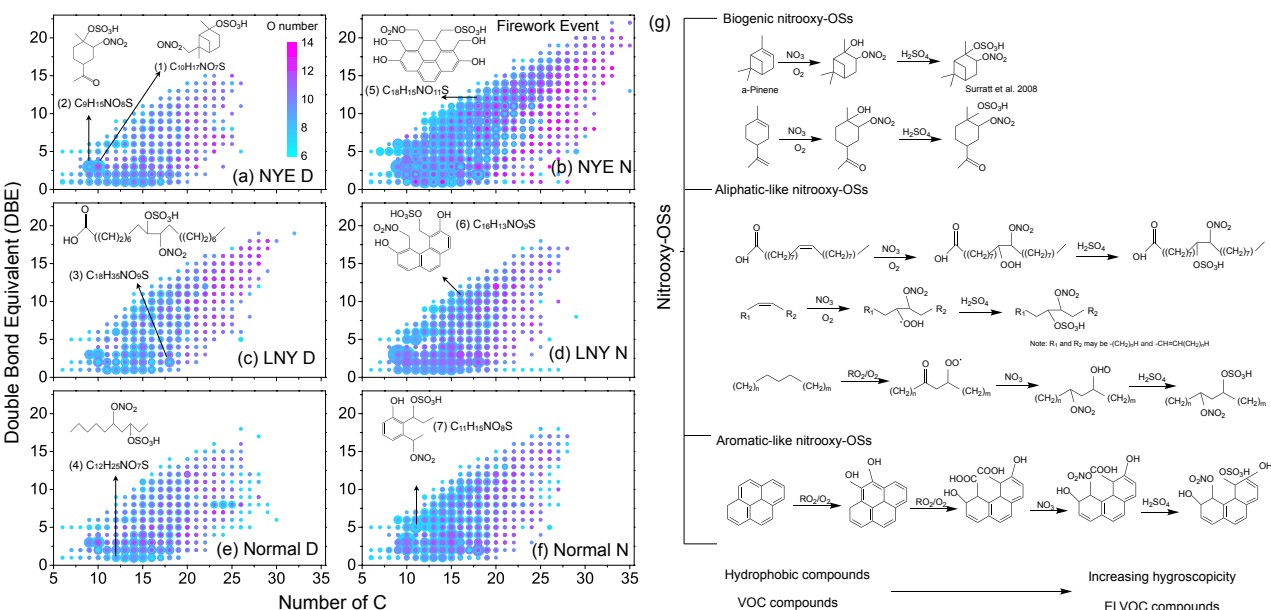

**Figure 4.** (a-f) DBE vs. C number for $N_1O_nS_1$ species. The color bar shows the number of O atoms. The size of the plots denotes the relative intensities of nitrooxy-OSs on a logarithmic scale. (1)–(7) are proposed structures of some nitrooxy-OSs, among which (1) and (2) have been reported previously (Surratt et al., 2008). Their relative intensities in each sample are shown in Table S3. (g) The proposed formation mechanisms of various group nitrooxy-OSs. The proposed mechanisms here are representative, but not determined.

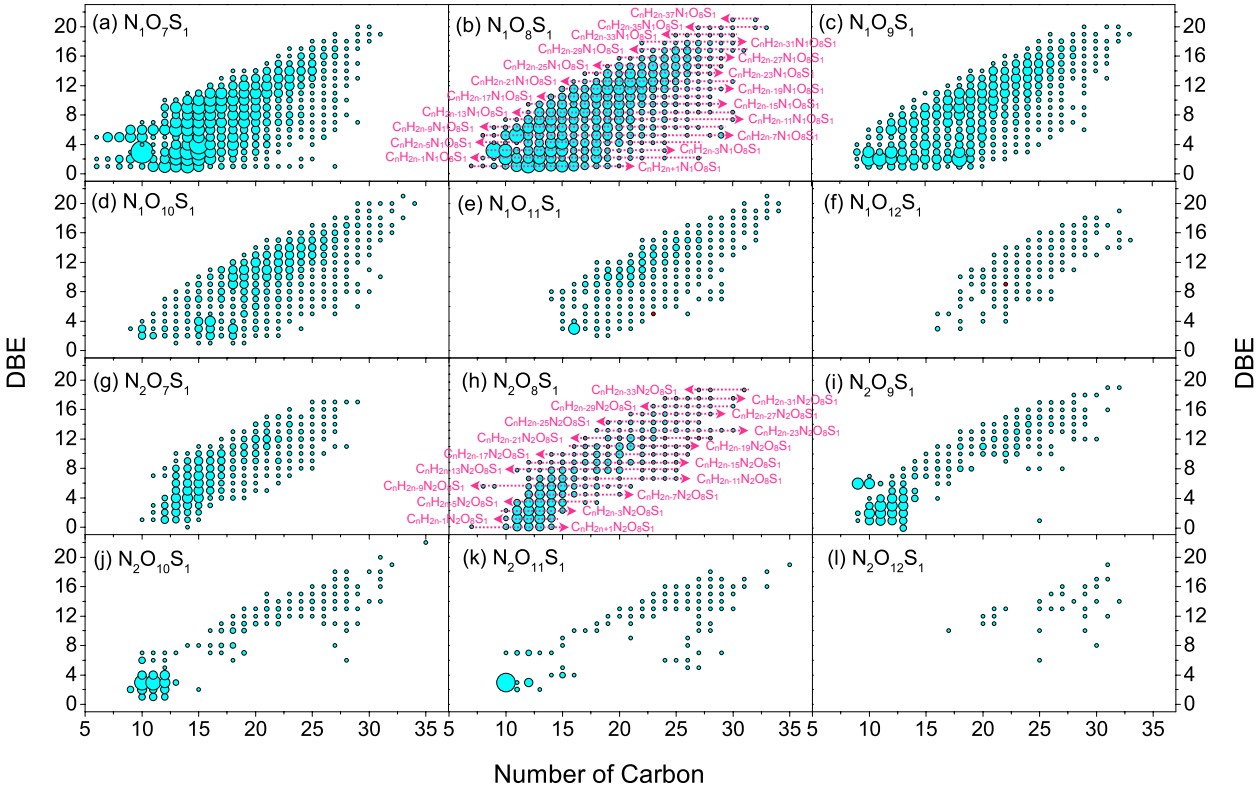

**Figure 5.** Molecular formulae distributions of $N_1O_7S_1$–$N_1O_{12}S_1$ and $N_2O_7S_1$–$N_2O_{12}S_1$ class species. The C and DBE number distributions of $N_1O_nS_1$ and $N_2O_nS_1$ class species in the Hazy D sample. The size of the symbols reflects the relative peak magnitudes of nitrooxy-OSs on a logarithmic scale. The pink arrow points and molecular formulae in $N_1O_8S_1$ and $N_2O_8S_1$ class species display the elemental composition of compounds as an example for all classes.

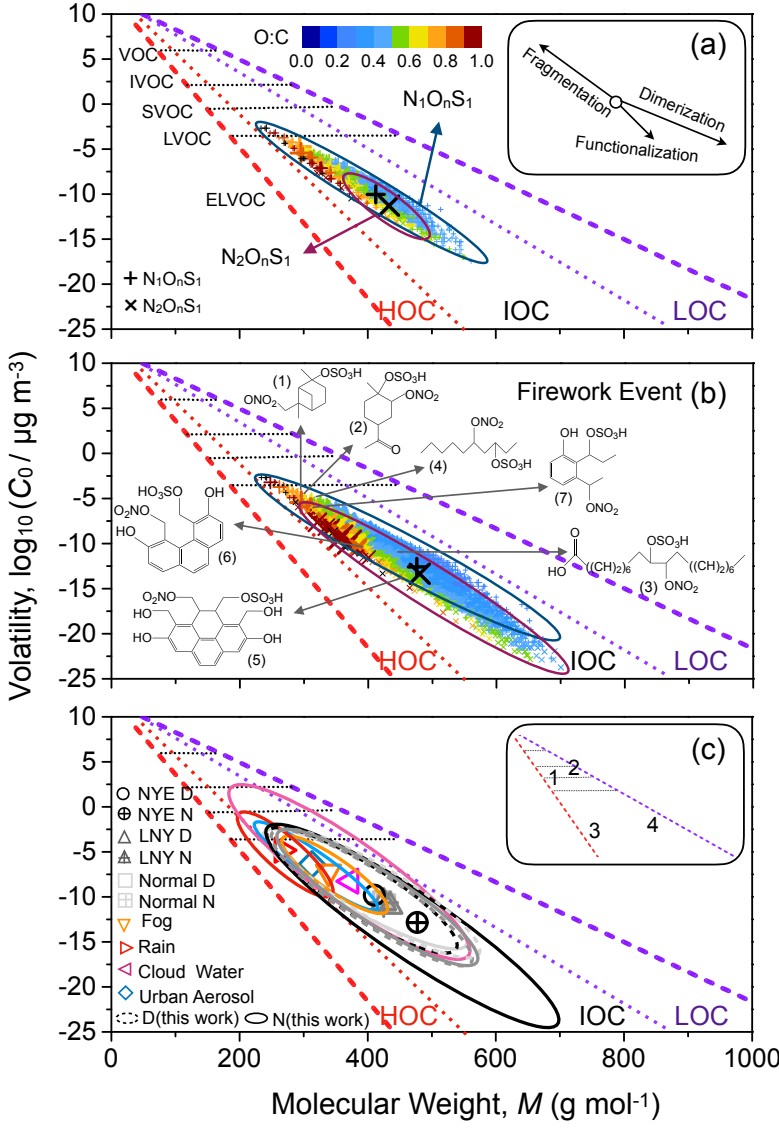

**Figure 6.** Molecular corridors and volatility characteristics for nitrooxy-OSs in (a) NYE D and (b) NYE N. (c) Comparison of nitrooxy-OSs in present work with those from urban aerosols (Lin et al., 2012; O'Brien et al., 2014), cloud water (Zhao et al., 2013), rain (Altieri et al., 2009), and fog (Mazzoleni et al., 2010). The top right of (c) shows the characteristic reaction pathways with most probable kinetic regimes (1. Aqueous-phase reaction; 2. Simple gas-phase oxidation; 3. Gas- or particle-phase autoxidation; and 4. Particle-phase dimerization) (Shiraiwa et al., 2014). The boundary lines denote sugar alcohols $C_nH_{2n+2}O_n$ with O/C = 1 (red) and linear n-alkanes $C_nH_{2n+2}$ with O/C = 0 (purple). The small plots denote the individual nitrooxy-OSs color-code by O/C ratio, and the larger ones show the surrogate nitrooxy-OSs with the average values of $M$, and $C_0$.