# Peer review of "Impact of firework on nitrooxy-organosulfates in urban aerosols during Chinese New Year Eve"

_Atmospheric Chemistry and Physics, 2021_

## Author Comment (AC1)

**Author Response to Comments of the Reviewers**

We appreciate the detailed and constructive comments and suggestions from the reviewers. The point-to-point responses to the comments are listed as below.

The *Reviewer comments are black italic font* and the Author responses are blue font.

**Author Responses to Reviewer #1**

*In this manicurist, Xie et al measured the nitrooxy-organosulfates (nitrooxy-OS) in the aerosols during Chinese New Year Eve and aimed at discussing the impact of firework on nitrooxy-OS formation. The paper is well written and in general intriguing. It provides unique information on the molecular characterization, classification and precursors of nitrooxy-OS in ambient aerosols during firework events. This information will help us better understand the formation pathways of nitrooxy-OS and importance of nighttime chemistry/aqueous chemistry under ambient conditions. Therefore, I recommend publication of this manuscript as long as the following concerns are properly addressed.*

**Response:** We really appreciate the valuable comments from the reviewer. We have made changes to both the main text and the supplemental information. Detailed responses are shown below.

*1. The title emphasizes "impact of firework", but there is no discussion on "firework" in the abstract part at all. It seems that "impact of firework" is not the whole story of this paper. Either of the title or the abstract should be revised accordingly for consistency of the paper.*

**Response:** The title has been revised to more consistent with the manuscript. Parts of the abstract have also been reformulated (on page 1 lines 1-2; on page 1 lines 20-25).

**"Increase of Nitrooxy-organosulfates in Firework-related Urban Aerosols during Chinese New Year Eve"**

**"High-molecular-weight nitrooxy-OSs with relatively low H/C and O/C ratios and high unsaturation are potentially aromatic-like nitrooxy-OSs. They considerably increased during the New Year's Eve that were affected by the firework emissions. We find that large quantities of carboxylic-rich alicyclic molecules possibly formed by nighttime**

**reactions. The sufficient abundance of aliphatic-like and aromatic-like nitrooxy-OSs in firework-related aerosols demonstrates that the anthropogenic volatile organic compounds are important precursors of urban secondary organic aerosols (SOA). Besides, more than 98% of those nitrooxy-OSs are extremely low-volatile organic compounds …"**

*2. Nighttime chemistry is one focus of this manuscript. Can the authors provide more information (e.g., meteorological condition, NOx/O3 concentrations, and if possible, VOC and aerosol chemical composition) to support their discussion on nighttime nitrooxy-OS formation during the focused time period?*

**Response:** The concentrations of chemical components in aerosols have been added in Table S1 in the supplemental information, including water-soluble organic nitrogen (WSON), and water-soluble $SO_4^{2-}$ and $NO_3^-$ (on page 4 lines 7-13).

**"They were consistent with the concentration trend of water-soluble organic nitrogen, which were significantly higher in the nighttime than that in the daytime, particularly in NYE N (Table S1) … Meanwhile, the heavy emissions of nitrogen oxide during the firework event could elevate the production rate of $NO_3$ radicals (Ljungström and Hallquist, 1996; Kiendler-Scharr et al., 2016), and previous study showed a good correlation between $NO_3$ and the total concentration of nitrooxy-OSs at the night (Nguyen et al., 2014)."**

*3. Although presented in the table and figures, little discussion is made on the comparison between LNY D and LNYN. It is suggested that a few sentences discussion is added to show the unique situation of firework during the NYE N.*

**Response:** The discussion about the molecular composition in LNY D and LNY N has been made in the manuscript (on page 4 lines 4-5; on page 5 lines 7-8).

**"…, similar for the comparison between LNY N (1113) and LNY D (1097) samples, which were in agreement with previous studies…"**

**"…, and 22 and 23 compounds in LNY D and LNY N, respectively."**

*4. From Figure 3, it can be concluded that all nitrooxy-OS categories are enhanced during NYE*

*N. So what category is driven by the firework emission, and what is mainly due to the enhancement of nighttime chemistry? The current discussion is not clear enough.*

**Response:** Thanks. Firework emissions had the most impact on the lignin-like nitrooxy-OSs. The increase of the intensity of carbohydrates-like nitrooxy-OSs is mainly due to the enhancement of nighttime chemistry. The presentation has been added to the manuscript (on page 6 lines 31-33)

**"All up, all nitrooxy-OS categories were enhanced in NYE N, particularly for the lignin-like nitrooxy-OSs. Moreover, the intensity of carbohydrates-like nitrooxy-OSs increased due to the enhancement of nighttime chemistry."**

**References:**

Kiendler-Scharr, A., Mensah, A. A., Friese, E., Topping, D., Nemitz, E., Prevot, A. S. H., Aijala, M., Allan, J., Canonaco, F., Canagaratna, M., Carbone, S., Crippa, M., Dall Osto, M., Day, D. A., De Carlo, P., Di Marco, C. F., Elbern, H., Eriksson, A., Freney, E., Hao, L., Herrmann, H., Hildebrandt, L., Hillamo, R., Jimenez, J. L., Laaksonen, A., McFiggans, G., Mohr, C., O'Dowd, C., Otjes, R., Ovadnevaite, J., Pandis, S. N., Poulain, L., Schlag, P., Sellegri, K., Swietlicki, E., Tiitta, P., Vermeulen, A., Wahner, A., Worsnop, D., and Wu, H. C.: Ubiquity of organic nitrates from nighttime chemistry in the European submicron aerosol, Geophys. Res. Lett., 43, 7735-7744, 10.1002/2016gl069239, 2016.

Ljungström, E., and Hallquist, M.: Nitrate radical formation rates in scandinavia, Atmos. Environ., 30, 2925-2932, 1996.

Nguyen, Q. T., Christensen, M. K., Cozzi, F., Zare, A., Hansen, A. M. K., Kristensen, K., Tulinius, T. E., Madsen, H. H., Christensen, J. H., Brandt, J., Massling, A., Nojgaard, J. K., and Glasius, M.: Understanding the anthropogenic influence on formation of biogenic secondary organic aerosols in Denmark via analysis of organosulfates and related oxidation products, Atmos. Chem. Phys., 14, 8961-8981, 10.5194/acp-14-8961-2014, 2014.

---

## Author Comment (AC2)

**Author Response to Comments of the Reviewers**

We appreciate the detailed and constructive comments and suggestions from the reviewers. The point-to-point responses to the comments are listed as below.

The *Reviewer comments are black italic font* and the Author responses are blue font.

**Author Responses to Reviewer #2**

*General comment:*

*This manuscript characterized nitrooxy-OSs in urban aerosol at a molecular level using Fourier transform ion cyclotron resonance mass spectrometry. The authors found that fireworks have substantial effects on nitrooxy-OS formation especially in nighttime and they provided significant information about the sources, classification, and physiochemical properties of nitrooxy-OSs. Overall, the paper is written well, and the results is of great importance for nitrooxy-OS study. However, the abstract and summary need to be improved. I recommend this paper to publish at Atmospheric Chemistry and Physics if the authors can account for the following comments.*

**Response:** We really appreciate the valuable comments from the reviewer. We have made changes to both the main text and the supplemental information. Detailed responses are shown below.

*Specific comments:*

*P1, lines 23-25. It is well known that SOA could be generated from the atmospheric oxidation of both anthropogenic and biogenic VOCs. This sentence should be rewritten to highlight the result of the current work. In addition, the authors should demonstrate the impacts of nighttime chemistry and firework on nitrooxy-OSs formation in the abstract to response the title of this manuscript.*

**Response:** Parts of the abstract have been reformulated (on page 1 lines 20-25).

**"High-molecular-weight nitrooxy-OSs with relatively low H/C and O/C ratios and high unsaturation are potentially aromatic-like nitrooxy-OSs. They considerably increased during the New Year's Eve that were affected by the firework emissions. We find that large quantities of carboxylic-rich alicyclic molecules possibly formed by nighttime reactions. The sufficient abundance of aliphatic-like and aromatic-like nitrooxy-OSs in**

**firework-related aerosols demonstrates that the anthropogenic volatile organic compounds are important precursors of urban secondary organic aerosols (SOA). Besides, more than 98% of those nitrooxy-OSs are extremely low-volatile organic compounds …"**

*P2, line 23. Please provide a definition about high-molecular-weight compounds.*

**Response:** The definition about high-molecular-weight compounds has been provided in the manuscript (on page 2 lines 23-24).

**"… particularly for high-molecular-weight (HMW, molecular weight more than 500 Da) compounds …"**

*P3, line 8. Please note the manufacturer and model of the aerosol sampler.*

**Response:** The detail information about of the aerosol sampler has been presented in the manuscript (on page 3 lines 7-9).

**"The total suspended particle (TSP) samples were collected on prebaked quartz filters (20 cm × 25 cm, Pallflex) using a high-volume air sampler (Kimoto, Japan), …"**

*P3, line 16. The authors highlighted the ultrahigh resolution of FT-ICR MS. Please show the specific value of MS resolution in the manuscript.*

**Response:** The specific value of FT-ICR MS resolution has been shown in the manuscript (on page 3 line 17).

**"An average resolving power (m/$\Delta$m50%) of over 400,000 at m/z ~400 was achieved."**

*P3, line 19-25. Please supply more information about the MS data analysis.*

**Response:** Some information about the MS data analysis has been provided in the manuscript. (on page 3 lines 19-25)

**"A molecular formula calculator was used to calculate formulas with elemental compositions up to 50 of $^{12}$C, 100 of $^{1}$H, 50 of $^{16}$O, 2 of $^{14}$N, and 1 of $^{32}$S atoms (Cao et al., 2016; Mazzoleni et al., 2012). Several conservative rules were used as further restrictions for the formula calculation (i.e. the elemental ratios of H/C < 2.5, O/C < 1.2, and S/C < 0.2, and the N rule for even electron ions) (Wozniak et al., 2008; Zhang et al., 2016). Unambiguous molecular formula assignment was determined with the help of the homologous series approach for multiple formula assignments (Koch et al., 2007;**

**Herzsprung et al., 2014). The isotopic peaks were removed in this study."**

*P4, line 2. As mentioned in the introduction, organonitrates that are more likely formed by nighttime chemistry instead of daytime reactions are important precursors for nitrooxy-OSs. Here, the results were derived from the combined influences of nighttime chemistry and fireworks. The focus of this study is to explore the impacts of fireworks on nitrooxy-OS formation. Why not analyze the particles (such as the sampler LNY N) collected in nighttime without fireworks?*

**Response:** We thank the reviewer for this comment. We have analyzed the aerosols in Normal N and LNY N which were collected in nighttime without fireworks. The comparison of molecular composition in aerosols between Normal N and NYE N can be found in the manuscript. For example, (on page 4 lines 30-31; on page 5 lines 7-9; on page 5 lines 26-28):

**"The average molecular weights rose from 411 ± 69 Da (Normal D) to 417 ± 78 Da (Normal N), and from 398 ± 69 Da (NYE D) to 449 ± 93 Da (NYE N)."**

**"From Table 1, 21 and 38 compounds with AI > 0.5 were observed in Normal D and Normal N, and 22 and 23 compounds in LNY D and LNY N respectively. Compared with 16 these compounds in NYE D, there were up to 83 compounds in NYE N."**

**"As for the most abundant CRAMs-like nitrooxy-OSs, they were more abundant during the nighttime (620 in Normal N) than daytime (375 in Normal D). However, NYE N contained about 1354 CRAMs-like compounds, which were three times more than NYE D …"**

*P4, line 9-10. A number of nitrooxy-OSs were only observed in sample NYE N, and these nitrooxy-OSs were suggested to be formed from the chemistry of fireworkderived precursors. Did primary nitrogen- and sulfur-containing compounds emitted directly from fireworks contribute to the high number of nitrooxy-OSs in NYE N?*

**Response:** Nitrooxy-OSs are a key component of secondary organic aerosols (SOA), and they substantially participate in the formation of SOA (Tolocka and Turpin, 2012; Ng et al., 2017; Bruggemann et al., 2020). Nitrooxy-OSs are primarily generated from both biogenic (Iinuma et al., 2007; Surratt et al., 2007; Gómez‐González et al., 2008; Surratt et al., 2008) and anthropogenic VOCs (Tao et al., 2014; Riva et al., 2015). Most of the high number of nitrooxy-OSs in NYE N are more likely to be secondary aerosols derived from special precursors. The

explanation of the nitrooxy-OSs can be found in the manuscript (on page 1 lines 33-34; on page 2 lines 5-6).

**"Nitrooxy-organosulfates (nitrooxy-OSs) … substantially participate in the formation of SOA … Nitrooxy-OSs can be generated from both biogenic (Iinuma et al., 2007; Surratt et al., 2007; Gómez‑González et al., 2008; Surratt et al., 2008) and anthropogenic VOCs (Tao et al., 2014; Riva et al., 2015)."**

*P6, lines 8-9. What does this sentence mean?*

**Response:** We are sorry for the confused presentation. To clarify, the confusing part were modified (on page 6 lines 16-18):

**"The intensity of carbohydrates-like nitrooxy-OSs was significantly higher than that of sulfur-free compounds reported in previous studies (Wozniak et al., 2008; Bianco et al., 2018). It is reasonable because …"**

*P6, lines 10-12. Please cite references.*

**Response:** The related references have been provided in the manuscript (on page 6 lines 19-20).

**"… which can be hydrolyzed to form polyhydroxy aldehydes or polyhydroxy ketones, tending to generate OSs and nitrooxy-OSs (Passananti et al., 2016; Ogino, 2021)."**

*Technical corrections:*
*P3, line 16. Electrospray ionization (ESI).*
**Response:** The 'ESI' has been defined (on page 3 line 16).
**"…the negative electrospray ionization (ESI) mode."**

*P7, line 20. S6(h).*
**Response:** Thank you very much for this suggestion. The 'S8(h)' has been corrected to 'S6(h)' (on page 7 line 30).
**"Figures S6(d) and S6(h) showed…"**

**References**

[revised manuscript text omitted]